# The Study of Forensically Important Insects Recovered from Human Corpses in Taiwan

**DOI:** 10.3390/insects14040346

**Published:** 2023-03-31

**Authors:** Wei-Lun Yan, Chiou-Herr Yang, Siew Hwa Tan, Chung-Yen Pai, Kan-Kun Li, Chen-Chou Chung

**Affiliations:** 1Department of Forensic Science, Central Police University, No. 56, Shuren Rd., Guishan Dist., Taoyuan City 333322, Taiwan; 2Forensic Science Center, Taoyuan Police Department, No. 3, Xianfu Rd., Taoyuan Dist., Taoyuan City 330206, Taiwan; 3Genetics and Molecular Biology, Institute of Biological Sciences, Faculty of Science, University of Malaya, Kuala Lumpur 50603, Malaysia; 4Department of Criminal Investigation, Central Police University, No. 56, Shuren Rd., Guishan Dist., Taoyuan City 333322, Taiwan; 5Forensic Science Center, Tainan Police Department, No. 3, Zhongzheng Rd., Xinying Dist., Tainan City 730207, Taiwan; 6Forensic Science Section, Hualien County Police Department, No. 21, Fuqian Rd., Hualien City 970018, Taiwan

**Keywords:** forensic entomology, human cadavers, calliphoridae, muscidae, sarcophagidae, Taiwan

## Abstract

**Simple Summary:**

The identification of insect species, larval development and succession sequencing on remains can provide information for estimating the minimum post-mortem interval (min-PMI). Many forensic entomology experiments using animal carcasses to study the insect’s activities have been reported for the past decades from different areas. Entomological data from human cadavers have been actively reported since the year 2000 from several countries. In this study, we reported and summarized the data of forensically important species associated with human corpses by collecting insect specimens from 114 forensic cases during 2011–2018 in Taiwan. Twenty-two species representing nine families were identified. Locations (indoor vs. outdoor), environments (urban vs. suburban), and temperatures are the important factors affecting insect communities. The fundamental data of case frequency and abundance of forensically significant insects recovered from human cadavers in Taiwan would be beneficial for forensic practitioners and criminal investigations.

**Abstract:**

A study of entomological specimens recovered from 117 human corpses in 114 forensic cases was conducted in Taiwan between 2011 and 2018. The comparisons and discussions of the entomological data were based on the locations (indoor vs. outdoor), environments (urban vs. suburban), season and decomposition stages of corpses. In the study, both morphology and DNA-based comparison methods were used to facilitate species identification. In total, nine families and twenty-two species were thus identified. The two most abundant fly species recovered from human corpses were *Chrysomya megacephala* (35.1%, 1735 out of 4949) and *Chrysomya rufifacies* (21.7%, 1072 out of 4949). As for case frequency, both the two were also the most common fly species (both 40%, 46 out of 114), particularly in outdoor cases (also both 74%, 25 out of 34). We found that *Chrysomya pinguis* and *Lucilia porphyrina* appeared in low temperature scenes in this study. *Synthesiomyia nudiseta* was the most common species detected on indoor (36%, 29 out of 80 cases) and urban (41%, 22 out of 54 cases) corpses. Sarcophagidae were strongly associated with urban environments (35%, 19 out of 54 cases), and *Parasarcophaga (Liosarcophaga) dux*, *Liopygia ruficornis* and *Boettcherisca peregrina* were the most frequent sarcophagid species collected from corpses. *Hydrotaea spinigera* was often found on corpses immersed in water (60%, three out of five cases) with advanced decay or remains stages. *Megaselia scalaris* was closely correlated with indoor cases (24%, 19 out of 80). In addition, *Piophila megastigmata* was collected from a corpse in the remains stage and the data represent the first report in Taiwan.

## 1. Introduction

Forensic entomology plays a significant role in criminal investigations. The most common application is the utilization of insects and other arthropods recovered from human cadavers to estimate the minimum post-mortem interval (min-PMI) for crime scene investigations [1,2]. The identification of insect species, larval development and succession sequence on remains can provide information for estimating the min-PMI.

Diptera and Coleoptera are two common orders found on corpses and hence they are the most-often used fauna in min-PMI estimation. According to Kulshrestha and Satpathy [3], Diptera plays an important role in the early stage of cadaver decomposition, whereas Coleoptera is useful in the late stage of decomposition. Calliphoridae is often the first family of flies to colonize a body after death. Other than Calliphoridae, species from Muscidae, Sarcophagidae, and Phoridae have also been recorded to be the useful flies in forensic cases [4].

Casework and research have mutually beneficial relationships in forensic entomology. Casework can provide challenges that need to be addressed with research [5]. Forensic entomology data from human cadavers have been actively reported since the year 2000 from Asian countries, such as Malaysia, Thailand and South Korea [6,7,8]. In Malaysia and Thailand, which have a tropical climate, *Chrysomya megacephala* and *Chrysomya rufifacies* were reported to be the predominant species from human remains [6,7]. In South Korea, which has a temperate climate, *Lucilia sericata* was found to be the most common species in human cadavers. Several forensic entomology experiments using animal carcasses have been reported from Taiwan, which revealed the insect activities and their distributions during different seasons, as well as separate stages of decomposition [9,10,11]. However, the forensic entomology data was only used once and reported in a single case associated with a human death in Taiwan thus far [12]. There is a need for more research and casework to be documented and integrated with forensically useful insect data in criminal investigations.

Accurate species identification of insect specimens collected from human cadavers is crucial for min-PMI estimation. Traditionally, species identification is carried out using morphological characteristics examination. However, it is notoriously difficult when dealing with immature specimens due to their undeveloped and similar morphological characteristics. The situation may become more complicated when the specimens collected are deteriorated in coping with difficult taxa, such as Sarcophagidae [13]. Therefore, after the first DNA-based identification report of insect species by Sperling et al. in 1994 [14], utilization of DNA sequencing, in particular, for the cytochrome oxidase subunit I gene, has become a popular alternative to facilitate species identification [15,16,17]. In this paper, both morphology and DNA-based identification methods were used for specimens recovered from human corpses. A list of forensically important dipteran species was obtained from Taiwan for the first time. This baseline data, including the DNA sequences obtained, would be useful for future analyses of forensic cases in Taiwan.

## 2. Materials and Methods

### 2.1. Sample Collection

Insect specimens were collected from human corpses by forensic investigators at the scenes. The procedure of each case was as follows:Photograph the scene and the corpse before insect collection.Observe the insects around the corpse and locate the place where they appeared.Sample the specimens with tweezers or spoons and keep them in small plastic bottles (mainly aiming at the part where the insects appeared or congregated. If there are more than two parts or the whole body covered with insects, then use random sampling method). There were no traps or baits used during the collection at the scene. Therefore, no live flies were caught, only a few dead adults were picked up.For laboratory analysis, all the specimens (except dead adults fly) were killed by 98 °C hot water for about 1 min and then stored in small glass bottles with 70% alcohol. Dead adult flies were preserved directly in 70% alcohol.Document information of forensic data (name, gender, age, location of corpses found, possible cause of death, decomposition stage of corpse) and climate data (according to nearby weather stations).

The case location and environment were categorized into indoor vs. outdoor and urban vs. suburban (including rural), respectively.

### 2.2. Morphology-Based Identification

For each case, the samples were examined under a stereomicroscope (M165C, Leica, Wetzlar, Germany). For eggs, first and second instars larvae and pupa specimens which were difficult to identify by the microscope were processed with DNA identification. For third instar larvae, the morphology identification was based on the method by Sukontason et al., Szpila, Velásquez et al., and Fan [18,19,20,21]. After morphology identification, one of these specimens was picked up for DNA analysis (e.g., Calliphoridae, Sarcophagidae and Muscidae depending on the needs). For adults, the beetles (*Dermestes maculatus* and *Necrobia rufipes*) were identified according to Almeida et al. [22]. The dead flies (*C. nigripes* and *C. megacephala*) were analyzed by DNA identification.

### 2.3. DNA-Based Identification

Total genomic DNA was extracted from each insect specimen. For immature specimens (egg, larvae and pupa), about 1 mm of the body of each specimen (larvae and pupa) or a small piece (egg) were used for DNA-based identification. Immature specimens were dried at 70 °C in an incubator prior to DNA extraction. For dead adult fly specimens, 1–2 legs were used for DNA extraction. Samples were ground into powder with a disposable plastic micropestle in 1.5 mL microcentrifuge tube, and DNA were extracted using QIAamp DNA Mini kit (Qiagen Inc., Germantown, MA, USA) according to the manufacturer’s instructions. The DNA obtained was used as the template for PCR using Phire Tissue Direct PCR Master Mix (Thermo Fisher Scientific, Waltham, MA, USA). Two DNA regions of mitochondrial *cytochrome oxidase subunit I* (*COI*) gene was amplified using primers C1-J-1718 (5′-GGAGGATTTGGAAATTGATTAGTTCC-3′) and TL2-N-3014 (5′-TCCAATGCACTAATCTGCCATATTA-3′) [23] or LCO1490 (5′-GGTCAACAAATCATAAAGATATTGG-3′) and HCO2198 (5′-TAAACTTCAGGGTGACCAAAAAATCA-3′) [24].

DNA sequence chromatograms were edited using BioEdit v.7.2.5 [25] and ContigExpress in the Vector NTI Advance v10.3 (Invitrogen, Waltham, MA, USA) to remove the sequences of primers and to eliminate discrepancies between overlapping sequence data. The COI sequences obtained were then compared to the Diptera sequences from the National Center for Biotechnology Information (NCBI) database by nucleotide Basic Local Alignment Search Tool (BLASTN) function for species identification with 99% similarity in BLASTN (except *Hermetia illucens* with 98%) was included in this study.

## 3. Results

A total of 117 human corpses (20 females and 97 males, aged from several days to 97 years old) were sampled from 114 crime scenes. There were three cases with two corpses found at the same scene. Fifty-four (54) cases occurred in urban areas and 60 cases in suburban areas, respectively. Corpses found in indoor locations (n = 80) were more frequent than corpses at outdoor locations (n = 34). Twenty-two species representing nine families were identified. Fanniidae sp. and Stratiomyidae sp. were classified by morphology identification only. The BLASTN search in NCBI database was not available for the DNA sequences of Phoridae sp. and Dermestidae sp. that less than <98% similarity was excluded from DNA-based identification. Decomposition stage of the corpse was established and divided into four stages during the autopsy was divided into four stages, namely the fresh (n = 6), bloated (n = 58), decay (n = 46) and remains (n = 7). In our work, we did not distinguish the active decay and advanced decay stages as defined by Payne [26], because these two stages are very difficult to differentiate clearly. Of all the 114 forensic cases, 55 were collected from Northern Taiwan, 35 from Southern Taiwan and 24 from Eastern Taiwan (Figure 1). The tropical and subtropical climate regions of Taiwan are separated by the Tropic of Cancer; where Northern Taiwan is classified as the subtropical climate whereas Southern Taiwan is grouped as the tropical climate. The information of each case and insect species was shown in Appendix A. 

The case frequency of fly species collected from different locations and environments is summarized in Table 1, representing the percentage of the corpses colonized with specific fly species. The indoor cases had a higher species diversity of forensically useful flies with 19 species, as compared to outdoor cases (11 species). Generally, Calliphoridae (58%, 66 out of 114), Muscidae (32%, 36 out of 114), and Sarcophagidae (23%, 26 out of 114) were the three high frequency families recovered from human corpses. Indoor and outdoor cases had different predominant fly species. For the outdoor cases, all of the corpses were colonized with Calliphoridae (100%, 34 out of 34), followed by Muscidae (15%, 5 out of 34) and Sarcophagidae (6%, 2 out of 34). In contrast, for the indoor cases, 40% (32 out of 80) of the corpses were colonized with Calliphoridae, 39% (31 out of 80) with Muscidae and 30% (24 out of 80) with Sarcophagidae. These results imply that the Calliphoridae emerged on both outdoor and indoor corpses, whereas families of Muscidae and Sarcophagidae arose on indoor corpses more than outdoor corpses. The two most common calliphoridae species collected from outdoor cases were *C. megacephala* (74%) and *C. rufifacies* (also 74%). Other minor Calliphoridae species involved in outdoor cases were *Chrysomya pinguis*, *Ceylonomyia nigripes*, *Hemipyrellia ligurriens* and *Lucilia porphyrina*. Interestingly, *Synthesiomyia nudiseta* was the most frequent species recovered from indoor corpses (36%) and was more predominant than *C. megacephala* and *C. rufifacies* (both 26%) in indoor cases. Almost one-third of the indoor cases were associated with *S. nudiseta,* indicating that it is an indicator species for indoor cases. On the contrary, *S. nudiseta* was rarely found from the outdoor corpses. Family Sarcophagidae including *Parasarcophaga dux*, *Liopygia ruficornis*, and *Boettcherisca peregrina* were more commonly collected from indoor cases (33%) than from outdoor cases (6%).

Our result showed that Calliphoridae had higher colonization frequency for suburban cases (82%) while Muscidae (43%) and Sarcophagidae (35%) were the high frequency fly families for urban cases. *Chrysomya megacephala* (52%) and *C. rufifacies* (55%) were the two most common species in suburban cases but *S. nudiseta* (41%) and Sarcophagidae species (35%) were more common in urban cases than in suburban cases.

The case frequency of fly species collected in different stages of human decomposition is presented in Table 2. Most of the corpses were in bloated (n = 58) and decay (n = 46) stages. The Calliphoridae were the most common family found in the bloated stage, followed by Sarcophagidae and Muscidae. *Chrysomya megacephala* was the most frequently collected species in the bloated stage, with the next frequently encountered species being the *C. rufifacies* and *S. nudiseta*. There were four species of Sarcophagidae (*P. dux*, *L. ruficornis*, *Parasarcophaga harpax* and *B. peregrina*) collected in the bloated stage of human decomposition. A total of sixteen species were recovered during the decay stage of decomposition. This represents the greatest species diversity among three human decomposition stages. In the decay stage, the Calliphoridae were the most common collected family, followed by Muscidae and Sarcophagidae. *Chrysomya megacephala* and *C. rufifacies* were the most common species, followed by *S. nudiseta* and *M. scalaris* in the decay stages.

The case frequency of fly species collected in separate seasons is shown in Table 3. Among the 114 cases, 41 occurred in the summer (Jun. to Aug.), followed by spring (Mar. to May., n = 32), winter (Dec. to Feb., n = 21) and fall (Sept. to Nov., n = 20). The temperature range was taken from the case that occurred in each season. There were ten common fly species that were present more than four times (n ≥ 5). *Chrysomya rufifacies* and *C. megacephala* were present in all seasons. *Chrysomya pinguis* was present in spring and winter and *Lucilia porphyrina* presented in spring and fall. *Synthesiomyia nudiseta* was present in all seasons, but less frequently in summer. *Hydrotaea spinigera* was present in fall and winter. Sarcophagidae appeared more frequently in summer than other seasons, especially *B. peregrina* and *P. dux*; however, *L. ruficornis* could be collected in winter. *Megaselia scalaris* was present in all seasons. 

Table 4 summarizes the rate of a predominant fly species found on corpses in all 114 cases. The predominant species is defined as one species whose population rate is the highest (usually more than 50%) among the insect samples that were collected from one case. The Calliphoridae were the predominant species (91% of the 34 cases) in outdoor cases, especially the *C. rufifacies* (41%) and *C. megacephala* (41%), followed by Muscidae (9%). In indoor cases, Calliphoridae were still the predominant species, but the predominant rate appeared to decline (30% of the 80 cases). However, the predominant rate of Muscidae (28% of the 80 cases), especially *S. nudiseta* (26%), Sarcophagidae (15%), and Phoridae (21%) all increased. This situation is similar to urban vs. suburban areas. In suburban cases, Calliphoridae often become the predominant species (73% of the 60 cases), especially *C. megacephala* (38%) and *C. rufifacies* (28%), followed by Muscidae (12%). In urban cases, Muscidae become the predominant species (33% of the 54 cases), especially *S. nudiseta* (31%), followed by Calliphoridae (20%), Sarcophagidae (20%), and Phoridae (20%). Our result also revealed that most of the forensic cases were colonized with one species (n = 52) and two species (n = 41). The insect species diversities (1 to 5) of indoor bodies colonization are higher than those of outdoor cases. These results are similar to those reported by Syamsa et al. [27].

Figure 2a shows the abundance of the insects colonizing on corpses in this study. A total of 4949 specimens (eggs not included) were collected from the 114 cases. Obviously, the most common species was *C. megacephala* (35.1%, 1735 out of 4949), followed by *C. rufifacies* (21.7%, 1072 out of 4949), *S. nudiseta* (16.5%, 819 out of 4949) and *M. scalaris* (8.6%, 424 out of 4949). The Figure 2b revealed the abundance of the insects on corpses between indoor and outdoor cases. The most common species for the indoor cases was *S. nudiseta* (28.5%, 788 out of 2769), followed by *C. megacephala* (22.6%, 626 out of 2769) and *M. scalaris* (15.3%, 424 out of 2769). For the outdoor cases, the most common species was *C. megacephala* (50.9%, 1109 out of 2180), followed by *C. rufifacies* (37.4%, 815 out of 2180). Figure 2c displays the insects’ abundance between urban and suburban areas. The most common species in urban area was *S. nudiseta* (28%, 551 out of 1969), followed by *C. megacephala* (24.1%, 475 out of 1969). For the suburban area, the most common species was *C. megacephala* (42.3%, 1260 out of 2980), followed by *C. rufifacies* (29.3%, 872 out of 2980).

## 4. Discussion

In this study, the forensically important species (species used for PMI estimation or indicator species for location or temperature) of forensic cases from 2011 to 2018 in Taiwan was summarized. During the course of the study, if the specimens collected from the crime scenes were in young immature stages, they posed a difficulty on the morphological identification. In such situations, DNA-based identification was used, in which more benefits can be obtained compared to morphology-based identification. This method has been proven to be useful in identifying life stages of insects and in some complicated species, particularly for sarcophagid species [28,29]. However, one limitation for this method was that specimens’ DNA sequence data obtained could be absent from the GenBank database (e.g., Phoridae sp. and Dermestidae sp. in this study), which could result in species unidentified or misidentified. Therefore, both morphology and DNA-based identification methods are necessary and equally important for forensic application. 

Temperature condition (e.g., seasons, latitude), location (outdoor vs. indoor) and environment (urban vs. suburban) are the three important factors affecting the insect diversity and activity on corpses. *Chrysomya pinguis* and *L. porphyrina* were the two important species that only recovered from human corpses in the suburban areas during spring, fall and winter (colder seasons) in Northern Taiwan. In this study, we found that two species were shown up on the corpse simultaneously in three cases which were under this cool season, especially in mountain areas. A similar finding had been reported by Lin et al. [11], revealing that *C. pinguis* and *L. porphyrina* could serve as an important indicator for PMI estimation and body displacement since the two species were only present in Northern Taiwan. However, the two could also be found in mountainous areas of Southern Taiwan (with tropical climate) where the ambient temperatures are low. In the study of Yang and Shiao [30], *C. pinguis* occupied in lower temperature either in seasons or areas, and therefore *C. pinguis* played a more important role than *C. megacephala* in such environments. Monum et al. [31] recently reported that *C. pinguis* and *L. porphyrina* were retrieved from a human cadaver in a mountain (1200 m) of Northern Thailand during winter (15–18 °C). These two species are forested species that inhabit cool areas of the mountain in tropical countries [32]. It is useful to know the fly bionomics and their distributions because this data may provide crucial information for location and time in forensic investigations. 

In our results, family Calliphoridae were the most common species (both the abundance and cases frequency) in Taiwan, especially for the *C. megacephala* (35.1% in abundance) and *C. rufifacies* (21.7% in abundance)*,* thus being the two most forensically important fly species for PMI estimation. Moreover, the two species (*C. megacephala* and *C. rufifacies*) were strongly associated with case frequency of outdoor (74%) and suburban (>50%). These results are in concordance with reports from other countries with similar climate conditions such as Malaysia, Thailand and Australia [6,7,33,34]. For the outdoor cases, *C. megacephala* was the predominant species during the bloated stage of decomposition and then was displaced by *C. rufifacies* during the decay stage. When *C. rufifacies* grow into third instar larvae, their invasive and predacious behaviors would express and prey on other larvae co-colonizing on the corpses, especially on *C. megacephala* [35]. This competitive advantage has therefore made them become the predominant species in the later stage of decomposition.

For family Muscidae, *S. nudiseta* has become increasingly important in forensic cases and has been intensively studied in recent years [6,7,36,37]. This study represents the comparison of *S. nudiseta* frequency involved in forensic cases in Taiwan with other adjacent countries. Results showed that the top three countries with the highest frequency of *S. nudiseta* associated with forensic cases were Taiwan (26%), Malaysia (16%) [6] and Thailand (10%) [7]. Nevertheless, *S. nudiseta* had not been collected in bait traps and mock experiments in Taiwan [9,10,11], so the importance of *S. nudiseta* may has been underestimated in the past years in Taiwan. The reason of the higher frequency (26%) of *S. nudiseta* in forensic cases from Taiwan compared to other countries, may be due to the higher numbers of our indoor cases (80 cases) since *S. nudiseta* was strongly associated with indoor cases.

We found that *S. nudiseta* was the most frequently encountered species (36%) in indoor cases and only collected in one outdoor case. A similar finding was reported by Sukontason et al. [7], in which *S. nudiseta* was collected from three indoor cases and none from outdoor cases in Northern Thailand. Lee et al. [38] also reported that *S. nudiseta* was only found on indoor corpses. A similar study in Europe by Velásquez et al. [36] reported 21 indoor forensic cases associated with *S. nudiseta* and only one associated to the outdoor case. In this particular outdoor case, the body was wrapped by a blanket, which might create a favorable condition for *S. nudiseta*. As shown in Figure 2a, *S. nudiseta* was the third most sampled species (16.5%, 819 out of 4949) in this study. However, it was the most common species collected in indoor cases (28.5%, 788 out of 2769) and urban areas (28%, 551 out of 1969) as shown in Figure 2b,c. *Synthesiomyia nudiseta* was not only a predominant species collected from indoors (26%, 21 out of 80) but it also had a higher occurrence frequency in urban (41%, 22 out of 54) than in suburban (13%, 8 out of 60) areas. These results are similar to the results reported by Kumara et al. [6] and Nazni et al. [39]. They inferred *S. nudiseta* as a eusynanthrophic species found in human dwelling places, especially indoors and this species is likely to occur in densely populated areas. Furthermore, *S. nudiseta* preferred and frequently appeared in cool and humid weather conditions, such as in spring and fall, although it was collected in all seasons in our study (Table 3).

Another forensically important Muscidae species in the study was *H. spinigera*. This species appeared five times, and were only obtained from suburban cases. These were found in the advanced decay or remains stage of decomposition and were usually found on the corpses immersed in the water (three out of five cases). *H. spinigera* could be obtained from mummified human cadavers, which was reported from forensic cases in Thailand and South Korea [8,40].

In this study, species from the family Sarcophagidae were strongly associated with indoor cases (30%, 24 out of 80) and were more present in the bloated stage (42%, 18 out of 43) than the decay stage (22%, 7 out of 32) of human decomposition, based on the third instar larvae of sarcophagids presented. Sarcophagids were reported to prefer shaded environments when colonizing corpses [7]. Our results corroborate with most of the forensic reports from other countries, in which most of the sarcophagid species were recovered from indoor corpses [6,7,8,41]. Sarcophagidae were also mainly found on dead bodies that occurred in urbanized areas [41]. This finding is congruent to our result that sarcophagid species occurred more frequently in urban cases (35%, 19 out of 54) than suburban cases (12%, 7 out of 60). Regarding seasonal effects, about 60% of the forensic cases involving Sarcophagidae were during summer (July and August) in Switzerland [41]. Similarly, all forensic cases with sarcophagid colonization were recovered during summer in South Korea [8]. Schroeder et al. [42] had even named *Sarcophaga* sp. as ‘summer species’ in their study. These indicated that sarcophagids are more likely to appear in a warmer atmosphere, which is in line with our finding that the sarcophagid samples were often collected during summer (Jun. to Aug.) at temperatures ranging from about 24 to 32 °C. In addition, we also found the occurrence frequency of sarcophagid on corpses in Southern Taiwan was higher than Northern Taiwan, where Southern Taiwan has a higher average ambient temperature than that in Northern Taiwan generally.

It is noteworthy to mention that among three sarcophagid species in Taiwan, *L. ruficornis* was the most common species found from human corpses. It is a synanthropic species which is widespread throughout the world. In our study, it was collected only from indoors (Table 1) and throughout the year but it was most active in the summer. *Liopygia ruficornis* was also reported to be the most important sarcophagid species from forensic cases that occurred in Thailand and Malaysia [6,7,29]. Another two sarcophagid species in Taiwan, *P. dux* and *B. peregrina*, also had been reported in the forensic cases of South Korea [8].

Family Phoridae, also known as scuttle fly, are an important fly species recovered from indoor corpses, as they could easily access concealed corpses or corpses in closed rooms due to its small body size [6,43,44]. Our study revealed that *M. scalaris* was only collected from indoor cases (25%). It could be due to the invasion impact by larvae of Calliphoridae while in outdoor cases. In Thailand, *M. scalaris* could be found in forests [7]. *Megaselia scalaris* was one of the predominant species from indoor cases other than *S. nudiseta* and *L. ruficornis*. The number of *M. scalaris* (15.3%, 424 out of 2769) was also the third highest in indoor cases, after *S. nudiseta* (28.5%, 788 out of 2769) and *C. megacephala* (22.6%, 626 out of 2769). As a result, *M. scalaris* could serve as an alternative for estimating the min-PMI for indoor cases when other flies, particularly flies of a larger size, were unable or delayed to access to an enclosed environment. In such circumstances, Phoridae species are remarkably important when involved in forensic cases with CO-poisoned and buried corpses, where corpses were usually found in sealed rooms and wrapped in bags.

The *P. megastigmata* was the only species of the family Piophilidae found in this study. The location of this case was an open area in a suburban area in Northern Taiwan. The corpse was in a car with windows closed (regarded as indoor) in winter. The possible cause of death was CO-poisoned (suicide). The corpse was in the remains stage with leather skin and fully clothed. The max-PMI could be more than two months according to the survey report. The larvae of *P. megastigmata* could be seen jumping during the collection at the scene. This data represents the first report of *P. megastigmata* found in human remains in Taiwan.

## 5. Conclusions

In criminal investigations, insect fauna found on a dead body is crucial in providing useful clues and evidence to solve the case. Based on the analyses of the 114 forensic entomology cases, *C. megacephala* and *C. rufifacies* were the two most abundant species in Taiwan, particularly in outdoor cases. While *S. nudiseta*, *M. scalaris* and Sarcophagidae were significant fly species recovered from indoor and urban cases. When a corpse was disposed in a closed space, Phoridae played an important role in min-PMI estimation. The *C. pinguis* and *L. porphyrina* may be suitable to serve as temperature indicator in forensic entomology in Taiwan, as they were only present during low temperature surroundings and within certain regions. The presence of *H. spinigera* indicated that the corpses may have been immersed in water. In this study, the larvae or pupa on/near the corpse were the main specimens collected, therefore, flying insects (e.g., fly, wasp) and ants might be overlooked. This study provides foundational data of forensically important insects recovered from human cadavers in Taiwan and would be useful for local forensic purposes.

## Figures and Tables

**Figure 1 insects-14-00346-f001:**
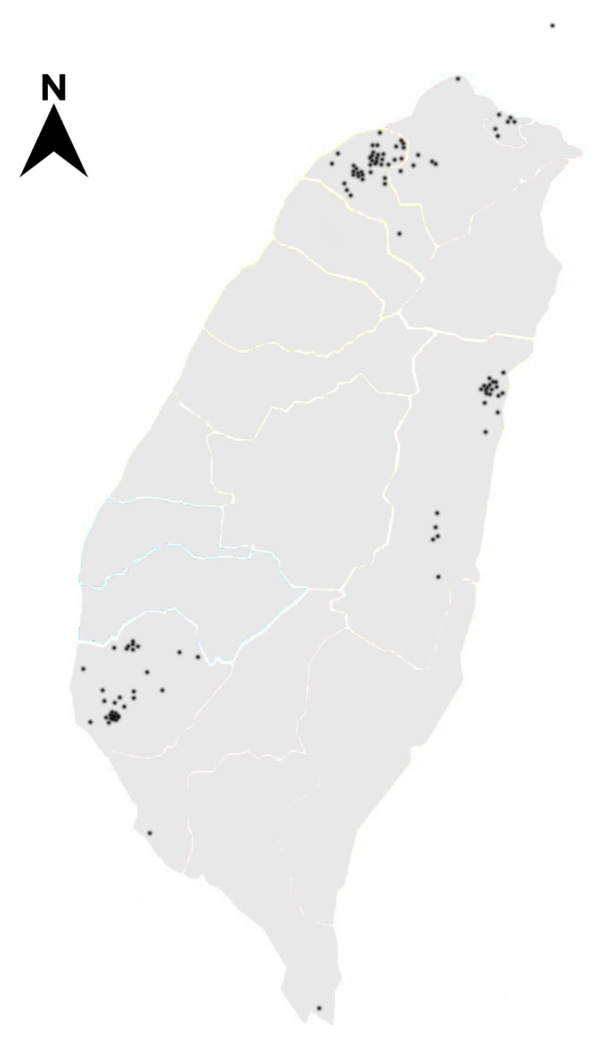
The locations of the 114 forensic cases in Taiwan (indicated by black spot).

**Figure 2 insects-14-00346-f002:**
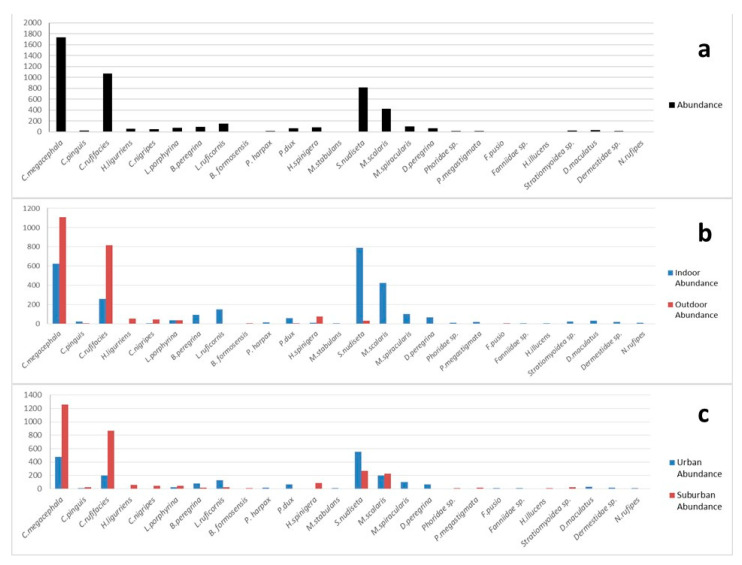
The abundance of the insects collected from the corpses in this study. (**a**) The total number and distribution of the specimens in the 114 cases (n = 4949); (**b**) comparison of species abundance on corpses between indoor (n = 2769) and outdoor (n = 2180) cases; (**c**) comparison of species abundance between urban (n = 1969) and suburban (n = 2980) areas.

**Table 1 insects-14-00346-t001:** Case frequency of fly species collected from 114 forensic cases in indoor/outdoor and urban/suburban in Taiwan from 2011 to 2018. (n, number of cases).

Family and Species	Total (n = 114)				
	Outdoor (n = 34)	Indoor (n = 80)		Urban (n = 54)	Suburban (n = 60)
	n, (%)	n, (%)		n, (%)	n, (%)
Calliphoridae	34 (100%)	32 (40%)		17 (31%)	49 (82%)
*Chrysomya rufifacies*	25 (74%)	21 (26%)		15 (28%)	31 (52%)
*Chrysomya megacephala*	25 (74%)	21 (26%)		13 (24%)	33 (55%)
*Chrysomya pinguis*	2 (6%)	3 (4%)		1 (2%)	4 (7%)
*Lucilia porphyrina*	2 (6%)	3 (4%)		1 (2%)	4 (7%)
*Ceylonomyia nigripes*	2 (6%)	1 (1%)		0	3 (5%)
*Hemipyrellia ligurriens*	1 (3%)	0		0	1 (2%)
Muscidae	5 (15%)	31 (39%)		23 (43%)	13 (22%)
*Synthesiomyia nudiseta*	1 (3%)	29 (36%)		22 (41%)	8 (13%)
*Muscina stabulans*	0	1 (1%)		1 (2%)	0
*Hydrotaea spinigera*	4 (12%)	1 (1%)		0	5 (8%)
Sarcophagidae	2 (6%)	24 (30%)		19 (35%)	7 (12%)
*Liopygia ruficornis*	0	11 (14%)		8 (15%)	3 (5%)
*Boettcherisca peregrina*	0	9 (11%)		6 (11%)	3 (5%)
*Parasarcophaga dux*	1 (3%)	6 (8%)		7 (13%)	0
*Boettcherisca formosensis*	1 (3%)	0		0	1 (2%)
*Parasarcophaga harpax*	0	1 (1%)		1 (2%)	0
Phoridae	0	23 (29%)		15 (28%)	8 (13%)
*Megaselia scalaris*	0	19 (24%)		11 (20%)	8 (13%)
*Megaselia spiracularis*	0	4 (5%)		4 (7%)	0
*Diplonevra peregrina*	0	1 (1%)		1 (2%)	0
Phoridae sp.	0	1 (1%)		0	1 (2%)
Fanniidae	1 (1%)	1 (1%)		2 (4%)	0
*Fannia pusio*	1 (3%)	0		1 (2%)	0
Fanniidae sp.	0	1 (1%)		1 (2%)	0
Piophilidae	0	1 (1%)		0	1 (1%)
*Piophila megastigmata*	0	1 (1%)		0	1 (2%)
Stratiomyidae	0	2 (3%)		0	2 (3%)
*Hermetia illucens*	0	1 (1%)		0	1 (2%)
Stratiomyidae sp.	0	1 (1%)		0	1 (2%)
Dermestidae	0	2 (3%)		2 (4%)	0
*Dermestes maculatus*	0	1 (1%)		1 (2%)	0
Dermestidae sp.	0	1 (1%)		1 (2%)	0
Cleridae	0	2 (3%)		2 (4%)	0
*Necrobia rufipes*	0	2 (3%)		2 (4%)	0

**Table 2 insects-14-00346-t002:** Case frequency of fly species collected in different stages of human decomposition from 114 forensic cases in Taiwan from 2011 to 2018. (n, number of corpse).

Family and Species	Stage of Decomposition (n = 117)			
	Fresh, n = 6	Bloated, n = 58	Decay, n = 46	Remains, n = 7
Calliphoridae				
*Chrysomya rufifacies*	1	18	25	2
*Chrysomya megacephala*	2	23	21	0
*Chrysomya pinguis*	1	0	4	0
*Lucilia porphyrina*	1	1	3	0
*Ceylonomyia nigripes*	0	0	2	1
*Hemipyrellia ligurriens*	1	0	0	0
Muscidae				
*Synthesiomyia nudiseta*	1	16	13	0
*Muscina stabulans*	0	0	1	0
*Hydrotaea spinigera*	0	0	2	3
Sarcophagidae				
*Liopygia ruficornis*	0	7	4	0
*Boettcherisca peregrina*	0	7	2	0
*Parasarcophaga dux*	0	5	2	0
*Boettcherisca* *formosensis*	0	0	1	0
*Parasarcophaga* *harpax*	0	1	0	0
Phoridae				
*Megaselia scalaris*	0	14	5	0
*Megaselia spiracularis*	0	3	1	0
*Diplonevra peregrina*	0	0	1	0
Phoridae sp.	0	1	0	0
Fanniidae				
*Fannia pusio*	0	0	1	0
Fanniidae sp.	0	1	0	0
Piophilidae				
*Piophila megastigmata*	0	0	0	1
Stratiomyidae				
*Hermetia illucens*	0	0	1	0
Stratiomyidae sp.	0	0	0	1
Dermestidae				
*Dermestes maculatus*	0	0	0	1
Dermestidae sp.	0	0	1	0
Cleridae				
*Necrobia rufipes*	0	0	0	2

**Table 3 insects-14-00346-t003:** Case frequency of insect species collected in different seasons. (n, number of cases).

Family and Species	Season (n=114)			
	Spring (16.2–29 °C) (n = 32)	Summer (23.8–31.8 °C) (n = 41)	Fall (11.4–32.4 °C) (n = 20)	Winter (13.4–23.6 °C) (n = 21)
Calliphoridae				
*Chrysomya rufifacies*	10	22	9	5
*Chrysomya megacephala*	12	24	6	4
*Chrysomya pinguis*	4	0	0	1
*Lucilia porphyrina*	4	0	1	0
*Ceylonomyia nigripes*	1	1	1	0
*Hemipyrellia ligurriens*	0	0	0	1
Muscidae				
*Synthesiomyia nudiseta*	13	5	7	5
*Muscina stabulans*	0	0	0	1
*Hydrotaea spinigera*	0	0	2	3
Sarcophagidae				
*Liopygia ruficornis*	1	6	1	3
*Boettcherisca peregrina*	2	5	2	0
*Parasarcophaga dux*	0	5	2	0
*Boettcherisca formosensis*	1	0	0	0
*Parasarcophaga harpax*	0	1	0	0
Phoridae				
*Megaselia scalaris*	6	4	6	4
*Megaselia spiracularis*	2	0	1	0
*Diplonevra peregrina*	1	0	0	0
Phoridae sp.	0	1	0	0
Fanniidae				
*Fannia pusio*	0	0	1	0
Fanniidae sp.	1	0	0	0
Piophilidae				
*Piophila megastigmata*	0	0	0	1
Stratiomyidae				
*Hermetia illucens*	0	0	1	0
Stratiomyidae sp.	0	0	0	1
Dermestidae				
*Dermestes maculatus*	1	0	0	0
Dermestidae sp.	0	1	0	0
Cleridae				
*Necrobia rufipes*	1	0	0	1

**Table 4 insects-14-00346-t004:** Rate comparison of the predominant species collected from the 114 forensic cases in indoor/outdoor and urban/suburban cases in Taiwan from 2011 to 2018. (n, number of cases).

Family and Species	Total (n = 114)				
	Outdoor (n = 34)	Indoor (n = 80)		Urban (n = 54)	Suburban (n = 60)
	n, (%)	n, (%)		n, (%)	n, (%)
Calliphoridae	31 (91%)	24 (30%)		11 (20%)	44 (73%)
*Chrysomya rufifacies*	14 (41%)	6 (8%)		3 (6%)	17 (28%)
*Chrysomya megacephala*	14 (41%)	16 (20%)		7 (13%)	23 (38%)
*Chrysomya pinguis*	1 (3%)	1 (1%)		0	2 (3%)
*Lucilia porphyrina*	1 (3%)	1 (1%)		1 (2%)	1 (2%)
*Ceylonomyia nigripes*	1 (3%)	0		0	1 (2%)
Muscidae	3 (9%)	22 (28%)		18 (33%)	7 (12%)
*Synthesiomyia nudiseta*	1 (3%)	21 (26%)		17 (31%)	5 (8%)
*Muscina stabulans*	0	1 (1%)		1 (2%)	0
*Hydrotaea spinigera*	2 (6%)	0		0	2 (3%)
Sarcophagidae	0	12 (15%)		11 (20%)	1 (2%)
*Liopygia ruficornis*	0	5 (6%)		4 (7%)	1 (2%)
*Boettcherisca peregrina*	0	3 (3%)		3 (6%)	0
*Parasarcophaga dux*	0	3 (3%)		3 (6%)	0
*Parasarcophaga harpax*	0	1 (1%)		1 (2%)	0
Phoridae	0	17 (21%)		11 (20%)	6 (10%)
*Megaselia scalaris*	0	12 (15%)		6 (11%)	6 (10%)
*Megaselia spiracularis*	0	4 (5%)		4 (7%)	0
*Diplonevra peregrina*	0	1 (1%)		1 (2%)	0
Piophilidae	0	1 (1%)		0	1 (2%)
*Piophila megastigmata*	0	1 (1%)		0	1 (2%)
Stratiomyidae	0	1 (1%)		0	1 (2%)
Stratiomyidae sp.	0	1 (1%)		0	1 (2%)
Dermestidae	0	2 (3%)		2 (4%)	0
*Dermestes maculatus*	0	1 (1%)		1 (2%)	0
Dermestidae sp.	0	1 (1%)		1 (2%)	0
Cleridae	0	1 (1%)		1 (2%)	0
*Necrobia rufipes*	0	1 (1%)		1 (2%)	0

## Data Availability

The data presented in this study are available from the corresponding author upon reasonable request.

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
