# Peer review of "The Study of Forensically Important Insects Recovered from Human Corpses in Taiwan"

_insects, 2023, doi:10.3390/insects14040346_

Round 1

Reviewer 1 Report

A useful baseline dataset to extend on to other developmental studies. Although this is readable, a grammar and phrasing check is required. 

Simple Summary and Abstract should contain data (%) rather than state 'usually', as this is vague. You state species are 'important' for different cases, but important in what way?

Intro has some phrasing/tone issues, but on the whole is fine

Methods: 

Line 100 - what is decedent personal data?

Line 103 - immatures/larvae or adults, or both? This is unclear. How were they collected/trapped/sampled? Reared or preserved?

Line 112 - so anything other than third instars were processed by DNA only? Or adults were morphologically identified too?

Line 115 - what is the centre part of the larva/pupa?

Line 117 - you mention adults here, but provide no collection or preservation details

Results

Table 2 - are these referring to cases with adults, larvae or both for each species? It would be good to see a split between which stages were collected in each case.

Discussion

Line 222 - what was the split between sample type, i.e. eggs, 1sts, 2nds, 3rds, pupae, adults? This would provide good information on how useful morphological ID is, as well as which species are visiting the cadavers vs which are colonising.

Line 229 - why could DNA not be obtained from degraded samples? Degraded in what way? This should be possible, you just might need smaller fragments amplified.

Line 239 - Ch. rufifacies must be in italics

Reviewer 2 Report

The manuscript contains new and very significant information to justify publication. The problem is actual and represents an important contribution to the knowledge of Diptera associated with human corps, representing a very significant progress for forensic research in Taiwan and other Asian countries.

So, I recommend it to be published as it is, with minor revision:

- modify the Summary Simple Summary to avoid repeated information;

- include accession numbers submitted to GenBank, from the performed molecular analysis;

- correct reference number 9: Formos. Entomol. 2003,

Reviewer 3 Report

Summary and Overview:

This study provides an overview of forensically important species associated with  human remains in Taiwan collected over a 6-year period. The authors provide a species list, and confirmed their morphological identifications with DNA sequencing. While this manuscript provides an overall species list of forensic insects in Taiwan, it does not provide any additional data or conclusions, nor does it include analyses to determine if there were statistical differences in the abundance or diversity of these insects based on factors such as season, location and habitat. The authors state that much more data was collected, but fails to report it or draw conclusions based on this data. The lack of abundance data in particular makes it difficult to draw any conclusions about how common these species actually area. Having only 1 specimen from a particular species, even if present in multiple scenes, does not mean that it is a primary colonizer on remains. There is no mention of the average temperature in the manuscript nor any data to indicate what life stage each species was collected in, so it’s difficult to make conclusions about colonization. If the authors include all of the additional data, perform proper analyses and clarify their findings, this manuscript would be suitable for submission. In it’s current form, this paper provides a species list and vague information about the forensic species in Taiwan.

This entire manuscript needs to be edited for English grammar, as that was a barrier to reading this paper.

Line Comments: I have specific comments and wording suggestions for each section below

Simple summary:

Line 22-23: “outdoor cases and suburban areas”

Line 24: “especially in urban areas”

Line 24: Remove line “It was an important species…overlooked in the past”

Line 26: “found strongly associated with urban areas”

Line 27-28: “are usually found in low temperature scenes”

Line 28: “Hydrotaea…is usually found on”

Line 30-31: “This study is expected to provide foundational data…useful for local forensic practitioners and”

Abstract:

Line 38: “the two most forensically”

Line 39-40: “they were found in all cases and were particularly abundant in outdoor cases”

Line 40: Remove “on the other hand” and edit sentence to “were found in low temperature scenes in this study”

Line 45: “Hydrotaea spinigera is often found on corpses immersed in water…”

Line 46: “Megaselia scalaris was closely associated with indoor cases”

Line 47: “Piophila…collexted from a corpse in the remains stage and hadn’t been previously reported.”

Introduction:

Line 54: “investigations”

Remove line 54-56 “it can provide…” since you explain mPMI in the next sentence

Line 58: “The identification of insect species, larval development and succession sequence on remains can provide information for estimating the mPMI”

Lines 63-64: Calliphoridae is often the first family of flies to colonize a body…

Line 64-65: remove sentence starting with “Calliphorid” since the reproductive rate and development are common across all blow flies and no specific to any species in Chrysomya or Calliphora genus.

Line 65-66: Calliphora are not the dominant species in forensic cases in general; if you are discussing Calliphora as primary colonizer in Taiwan, stat that, but otherwise, this sentence is not accurate

Line 70: “casework can provide challenges that need to be addressed with research”

Line 74: “have a tropical climate” and you need a reference for papers citing that C. megacephala and C. rufifacies are most common

Line 82-83: “There is a need for more research and casework to document forensically important insects in criminal investigations”

Methods:

Were any specimens kept alive for rearing to adult or awere all specimens killed? Were specimens hot water killed and preserved on scene or once they arrived to the lab? The methods here are very vague: what kind of bottles were used, was any food source provided, where on the remains were the insects collected from? Was there any standardization used for collecting at each scene? Were the investigators trained in any way before they collected from the scenes? Did the investigators know what different types of arthropods to observe?

Which keys were used to identify beetles, adult flies, other families of insects? Most of the references listed here are keys for 3rd instar larvae

Results:

Lines 135-137: very confusing the way this is worded; “In 3 cases, there were two sets of remains in the same scene and were colonized by the same species.” 

Line 146: maybe a map showing the locations of the scenes would be helpful?

Were you only focusing on flies and neglecting other insets or were there no beetles, wasps, ants or other arthropods present?

Lines 151-160: not sure this paragraph is effective, since you have the species list in a table below; this is redundant. Maybe just list the families and most abundant species here instead

Line 162: “the indoor cases had a higher species diversity”

Line 167: “In contrast, for indoor cases, 40%...”

Line 179: played important roles isn’t really relevant; they were present, that doesn’t’ mean they were important necessarily; do you mean played an important role in consuming the tissue?

Were there any cases that took place in a rural location?

What life stages were each species collected in? The auithors mentioned that they had eggs, larvae, pupae, puparia, and adults; where is this data?

Who did the morophological Sarcophagidae identification and what keys were used for this?

Table 3: I don’t think this table needs to be included if you discuss it in the paragraph above it; it’s not really contributing any additional information 

Missing results on weather, season, cause of death; all of this data is listed in the methods, but is not provided or discussed in the results. It would be great to see seasonal and climate related trends in forensically relevant insects

There are no statistical analyses to determine if there was a different in diversity/abundance based on habitat or location

What was the abundance of the insects collected? All of the data provided just has the number of cases that a species was found, but doesn’t tell me which ones were most abundant at these scenes. 

What was the mean temperature and relative humidity in each season in the urban v. suburban locations? The authors don’t report any temperature data or environmental data, which is crucial for insect development and arrival.

Discussion

Line 220: it’s not much of an analysis since there were no statistics; this is just a summary of forensically important species in this area

Line 268: “A similar finding was…”

Line 271: “In Europe…”

Line 292: Sarcophagids colonized in the bloated stage? Is this based on the presence of eggs when collected? This isn’t clear in this manuscript

Line 303: where in the results is there any information about the season an temperature for the insects recovered?

Reviewer 4 Report

The article addresses an interesting topic with practical application in
Forensic Entomology,
the insects recovered from human corpses in
Taiwan during 2011 – 2018
.  The article is generally well written, but minor corrections are needed:   - line 21: insect communities, because they are not social insects; - line 40: ...were usually found; - lines 41 – 42: ... was the most common species found on indoor and
urban corpses.
- line 45: ...was usually found; - line 56: utilization of insects; - line 59: fauna succession is related to max-PMI;   Several scientific names (scientific names of genera) appear not in italics
(lines 64, 65 and 302).
- line 109: species of immature specimens? Because of references 18-20? - line 136: with the same; - lines 142-143 and on table 2: total of 117 cases. Which corpses presented
more than on stage of decomposition?
- lines 164, 167, 170, 179, 182 and 331: Sarcophagidae; - line 170: families Muscidae and... - line 181: higher colonization frequency; - lines 184, 302, 304 and 308: sarcophagid; - line 211: which are similar to... - line 214: ...higher frequency for all diversities of fly species (1 to 5)...
than outdoor cases.
- line 220: forensic cases from 2011 to 2018 in Taiwan; - line 248: their result? - line 249: indicator for PMI estimation and body displacement; - line 272: only one associated to... - line 302: indicated that sarcophagids... - line 318: was only collected; - line 320: They? The calliphorids? - line 336: indicated the corpses that...

Round 2

Reviewer 3 Report

The authors still don't report the abundance of the insects, just the percentages. The percentages don't indicate much more information than what was previously reported. Abundance data would allow the authors to make conclusions with their work.
